



# Statistics of vertical velocities in supercooled cloud layers over Leipzig and Praia measured with Doppler lidar

Johannes Bühl[1], Patric Seifert[1], Ronny Engelmann[1], Julia Fruntke[2], and Albert Ansmann[1]

[1]Leibniz Institute for Tropospheric Research (TROPOS), Permoserstr. 15, 04318 Leipzig, Germany
[2]Deutscher Wetterdienst (DWD), Frankfurter Straße 135, 63067 Offenbach, Germany

*Correspondence to:* Johannes Bühl (buehl@tropos.de)

**Abstract.** This study presents statistics of vertical air velocity at the bases of supercooled shallow cloud layers separately for mixed-phase and liquid-only clouds. For the first time, this statistics is compared for clouds observed over a sub-tropical site at Cape Verde ($14.9°$ N, $26°$ W) and a mid-latitudinal site at Leipzig, Germany ($51.3°$ N, $12.4°$ E). Phase properties and spatio-temporal

5   extent of the cloud layers were obtained from combined observations with Doppler lidar, Raman polarization lidar, and cloud radar. The statistical properties of the vertical-velocity distributions in both mixed-phase and pure-liquid cloud layers are found to be similar at both measurement sites. Standard deviation of the vertical velocities at both sites was found to be $0.4\,\mathrm{m\,s^{-1}}$ and was also the same in pure-liquid and mixed-phase layers. Skewness groups around -0.4 for both sites, pointing

10   to radiative cooling as the driver for the cloud turbulence. Occasionally, positive skewness in some cloud layers indicated external drivers, e.g., gravity waves, for the turbulence. From the observed similarity in the vertical-velocity statistics derived at the base of supercooled liquid cloud layers at Praia and Leipzig it can be concluded that other factors besides cloud dynamics are responsible for the differences in ice formation efficiency reported previously for both sites.





## 1 Introduction

Interactions between aerosols, clouds and atmospheric dynamics are key process in the hydrological cycle on Earth and major unknowns in numerical climate projections (Boucher et al., 2014). Understanding this complex interaction has become a major challenge of current weather and climate research. One of the many open questions is, to what extent ice formation in clouds is influenced by aerosols and which role vertical air velocity plays in this context. Measurements of Kanitz et al. (2011) have shown strong differences in efficiency of heterogeneous ice formation in cloud layers formed under relatively clean conditions in the southern midlatitudes and those formed under rather polluted conditions in the northern midlatitudes. It is the goal of this work to present a statistics of vertical velocity in mixed-phase cloud layers observed at Praia (Cabo Verde) and Leipzig (Germany) with the future goal to disentangle the influence of aerosol and dynamics in these cloud layers.

Shallow mixed-phase cloud layers (altocumulus, stratocumulus) have been used repeatedly for the study of the interaction between aerosols, cloud droplets, and in-cloud dynamics on the basis of in-situ or remote-sensing measurements (Fleishauer et al., 2002; Ansmann et al., 2009; Tonttila et al., 2011; Zhang et al., 2010a; Bühl et al., 2016) and extensive modeling studies (Korolev and Field, 2008; Pinsky et al., 2015). Their potential impact on Earth's climate has recently been assessed by Bourgeois et al. (2016). Shallow mid-level cloud layer have the advantage that cloud-top temperature can be determined easily and they are accessible with methods of active remote sensing. The number of ice formation processes taking place within these clouds is usually limited to immersion freezing (Ansmann et al., 2009). Evidence that cloud ice forms via the liquid phase was also presented by de Boer et al. (2009) and Westbrook and Illingworth (2011). Recently, Schmidt et al. (2015) found a strong and direct coupling between aerosols at cloud base and cloud droplets indicating that background aerosol conditions play a major role in the formation and evolution of shallow cloud layers. Kanitz et al. (2011), Zhang et al. (2012) and Seifert et al. (2015) showed that temperature-dependent efficiency of ice formation in such cloud layers varies strongly by region and by the dominant type of aerosol present at cloud level. The sketch in Fig. 1 shows the temperature dependency of ice formation found by Kanitz et al. (2011) for Leipzig (Germany) and Punta Arenas (Chile). The temperature level at which 50% of all cloud layers produce ice is found to differ by 17 K between both sites. A very low ice-formation efficiency was also found at Praia (Cape Verde) (Ansmann et al., 2009). At Leipzig a much larger aerosol forcing is found than at Punta Arenas (Hamilton et al., 2014). Also at Praia, supercooled clouds were only found at heights well above the dust- and smoke-laden aerosol layers which are usually limited to heights up to 5 km in northern-hemispheric winter (Tesche et al., 2011b). Zhang et al. (2010b) found similar differences between mid-latitudinal and the tropical cloud layers. The question arises, if this apparent effect of aerosols on ice formation could also be attributed to differences in vertical wind statistics within mid-latitudinal and tropical cloud layers. Tonttila et al. (2011) derived vertical velocities in shallow cumulus clouds with a cloud





radar and used it in order to come up with a sub-grid parameterization of vertical velocity for a climate model (Tonttila et al., 2013).

We present in this work, for the first time, comprehensive statistics of vertical motions in cloud layers over one mid-latitudinal (Leipzig) and one sub-tropical site (Praia). Combination of Dopper lidar and cloud radar (Leipzig) or Doppler lidar and Raman lidar (Praia) are used to discriminate between precipitating and non-precipitating layers, as demonstrated in (Bühl et al., 2013). Observations of vertical velocities together with such a sophisticated discrimination between liquid and mixed-phase have not been presented in the literature before. Since a Doppler lidar is much less sensitive on the influence of falling droplets than a cloud radar, this combination allows the unbiased study of vertical velocities at cloud base, even for ice-producing clouds.

The paper is structured as follows. In Section 2 the datasets are described. The methodology of identification, selection and classification of cloud layers is presented in Section 3. The comparison of the datasets between Leipzig and Praia are done in Section 4. In this section, also the effect of integration times on the statistics of vertical velocity is explored, which might facilitate comparison with other datasets.

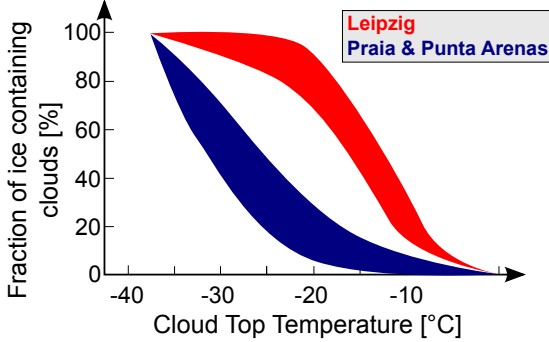

**Figure 1.** Distribution of ice-containing clouds (altocumulus, stratocumulus and cirrus) with temperature for Leipzig (red) and Praia or Punta Arenas (blue) adopted from Kanitz et al. (2011). The trends for for Praia and Punta Arenas can be considered equal within their corresponding uncertainty ranges.

## 2 Dataset

In this study, data from the second SAharan Mineral dUst experiMent (SAMUM-2) at Praia and the Up- and downdraft in Drop and Ice Nucleation Experiment (UDINE) at Leipzig are evaluated. SAMUM-2 was conducted in 2008. A lidar-observation site at Praia International Airport, Praia, Cabo Verde (14.9° N, 23.5° W) was established and continuous measurements with multiwavelength Raman/polarization lidar and Doppler lidar WiLi (Engelmann et al., 2008; Bühl et al., 2012) were performed in January, February and June 2008. An overview about the campaign can be found in



Ansmann et al. (2011), a detailed description of the involved lidar systems is give by Tesche et al. (2011a).

From 2010 to 2013, the measurement campaign UDINE (Up- and downdrafts in Drop and Ice Nucleation Experiment) was carried out at TROPOS, Leipzig ($51.3°$ N, $12.4°$ E). There, the remote-sensing measurement platform LACROS (Leipzig Aerosol and Cloud Remote Observations System, Wandinger (2012); Bühl et al. (2016)) provides continuous measurements with a combination of Raman lidar Polly[XT] (Engelmann et al., 2015), MIRA-35 cloud radar (Görsdorf et al., 2015), ceilometer, and microwave radiometer. During the UDINE campaign, measurements of LACROS were accompanied by vertical velocity observations of Doppler-lidar WiLi.

The main uncertainty of the vertical-velocity measurements with WiLi is pointing accuracy. If the beam is not pointed perfectly vertically, vector components of the horizontal wind speed are introduced into to the vertical velocity. With a precision scale, the vertical pointing of the system can be adjusted to better than $0.1°$, resulting in a maximum pointing error of about $0.05\,\mathrm{m\,s^{-1}}$ at a horizontal wind velocity of $30\,\mathrm{m\,s^{-1}}$. All measurements of WiLi were corrected for chirp influence with the method from Bühl et al. (2012). This correction is required for any Doppler lidar with a chirped pulse when analyzing vertical velocity at cloud bases, since chirp artifacts are most prominent at strong signal gradients. WiLi was run with a height-resolution of $75\,\mathrm{m}$ and a sampling time of $5\,s$ at Praia and $2\,s$ at Leipzig. For the comparison of vertical velocity statistics in the current work, the Leipzig dataset was interpolated to $4\,s$ integration time. Effects of integration time on the measured statistical values are discussed in detail in Section 4.2.

For both campaigns, meteorological data of the GDAS1 dataset provided by the Global Data Assimilations System (Kanamitsu (1989), web access: http://www.ready.noaa.gov/gdas1.php) has been used in order to derive profiles of horizontal wind velocity, wind direction and temperature for each cloud case.

## 3 Methods

### 3.1 Cloud identification and selection

From the datasets measured during the campaigns at Leipzig (UDINE) and Praia (SAMUM-2), a number of supercooled cloud layers (with a cloud-top temperature CTT $< 0\,°\mathrm{C}$) were selected. The selection process is described in Bühl et al. (2013) and comprise, e.g., small variations in cloud-top height (CTH), time-length of more than $15\,\mathrm{min}$, a geometrical cloud thickness smaller than $380\,\mathrm{m}$, a standard-deviation of CTH of less than $150\,\mathrm{m}$ over the whole cloud case and absence of ice particle seeding from above. 205 cases meeting these requirements were acquired with a combination of cloud radar and WiLi during UDINE. From the SAMUM-2 campaign, 70 cloud layers were selected. An example of a selected mixed-phase cloud layer is presented in Fig. 2. From that figure, first, the differences in sensitivities between lidars and radar become visible. Ceilometer and the Doppler lidar





mostly sense the liquid droplets at the base of the supercooled cloud layer, where the lidar signals
are then quickly attenuated. The cloud radar is most sensitive to the ice particles that are generated
at cloud top and consecutively falling down until evaporating at a height of about $4.5\,\mathrm{km}$. The cloud
is hence composed of two main parts: The mixed-phase cloud top ($5.0$ to $5.3\,\mathrm{km}$) where liquid water
particles and ice crystals coexist and the part below $5.0\,\mathrm{km}$ height with falling ice particles only,
which is usually referred to as "virga".

### 3.2 Cloud classification

Following identification and selection, the measured cloud layers are separated into ice-producing
*mixed-phase* and non-precipitating *liquid-only* cases by the method presented in Bühl et al. (2013).
The method relies on the measurement of the properties of the particles falling below cloud base.
If analysis of lidar/radar depolarization, fall velocity, and particle orientation points towards the
presence of falling ice particles and no seeding is detected from above the cloud layer, it can be
safely assumed that the ice has been produced within the stable, predominantly liquid cloud top
layer.

For both the SAMUM-2 and the UDINE campaign, lidar depolarization measurements were used
to distinguish between precipitating ice and water particles below the cloud layers and, thus, between
liquid and mixed-phase clouds. The depolarization lidar has a considerably higher particle detection
threshold than the WiLi lidar and can therefore sensitively detect all falling particles that potentially
influence the vertical velocity measurements. For the UDINE campaign additionally cloud-radar
measurements were used in order to identify falling particles below the stratiform cloud layers. For
each cloud layer, the CTH is derived from cloud radar or Raman lidar measurements. At this height,
CTT, horizontal wind speed, and horizontal wind direction are derived from the GDAS dataset.

### 3.3 Measuring vertical velocity at cloud base

For all identified cloud cases of both campaigns the vertical velocity statistics at the base of the
cloud-top layer is derived from measurements of Doppler lidar WiLi. In the vicinity of the cloud
base identified by Raman lidar or ceilometer, the maximum signal strength in the measurements of
WiLi is searched. The position of this maximum is then considered as the height of cloud base, where
the laser pulse is scattered mainly by liquid water droplets. The signal from these droplets dominates,
e.g., signals returned from ice particles present at this height. This effect can be seen in Fig. 2. In the
plot of vertical velocity (Fig.2b) only some scattered signals from – probably horizontally oriented –
ice crystals can be seen below cloud base. These oriented ice crystals cause specular reflection of the
lidar pulse, strongly increasing the backscatter signal. Turbulence within the cloud top layer prevents
ice particles from orienting towards their fall direction. Therefore, the signal from the small droplets
at cloud base is an undisturbed signal of vertical velocity and influence of falling ice particles is
negligible. In Fig. 2b, the histogram at cloud base of the example case is shown (inlay).





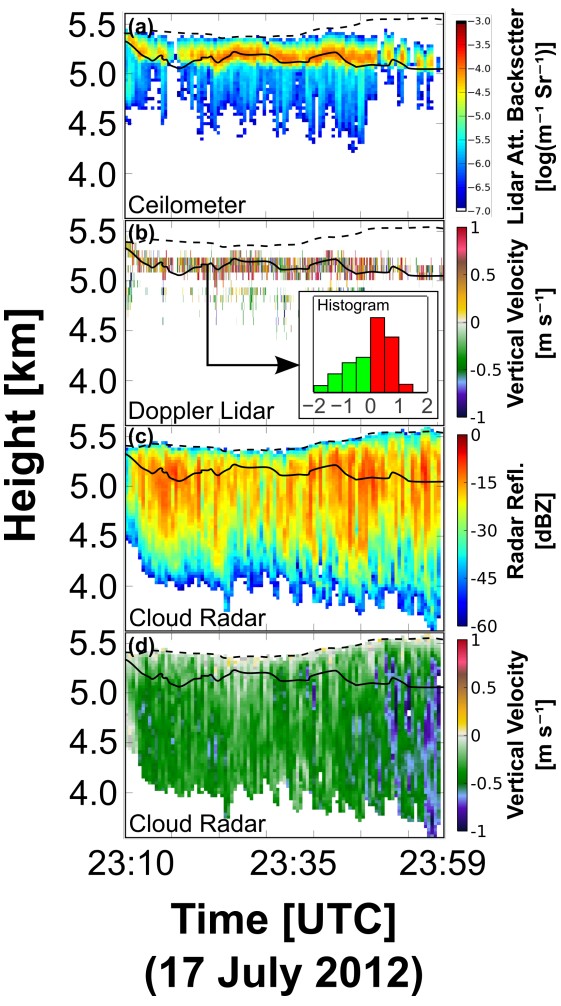

**Figure 2.** Mixed-phase cloud measured over Leipzig on 17 July 2012 between 23:10 and 23:59 UTC. Base and top of the mixed-phase cloud layer are marked with a solid and dashed line, respectively. Mean cloud-top temperature (CTT) is $-13°$C. Attenuated backscatter coefficient (a), Doppler-lidar vertical velocity (b), cloud-radar reflectivity factor (c) and cloud-radar Doppler velocity (d) are shown. Histogram in (b) comprises values of Doppler-lidar vertical velocity derived at cloud base.



### 3.4 Computing vertical-velocity statistics

For each cloud case, a histogram of vertical velocity is computed with a bin size of $0.2\,\mathrm{m\,s^{-1}}$. From
the normalized histogram with value $p_i$ in each interval the distribution moments $m_n$ are calculated:

$$m_n = \sum_i p_i (v_i - \bar{v})^n, \tag{1}$$

with $\bar{v}$ the mean vertical velocity for each cloud case, $n > 1$ and $i$ running over the number of
vertical-velocity intervals. From these moments, the standard deviation $\mathrm{STD} = \sqrt{m_2}$, sample skew-
ness $\mathrm{SKEW} = m_3/m_2^{3/2}$ and sample excess kurtosis $\mathrm{KURT} = m_4/m_2^2 - 3$ are computed. By design,
the statistical properties SKEW and KURT are susceptible to values which lie outside the normal
distribution. Hence, they react strongly on outlying values or technical noise. Therefore, all distri-
bution moments are calculated for the interval between $-2$ and $2\,\mathrm{m\,s^{-1}}$. As it is shown in Section 4,
this includes more than $99\%$ of all vertical-velocity values derived from cloud bases.

## 4 Results

### 4.1 Case-averaged statistics

Figure 3 shows the average properties of the airmasses into which the single cloud cases are embed-
ded. All values were derived for each cloud case at CTH from the GDAS1 dataset. CTT is physically
constrained and must lie between 233 and $273\,\mathrm{K}$ for supercooled mixed-phase cloud layers. From
the CTT histograms it is, however, visible that ice formation is more efficient at Leipzig, where
mixed-phase (i.e. ice-producing) clouds are clearly dominating below $260\,\mathrm{K}$. Average CTH is about
$3\,\mathrm{km}$ larger at Praia, which is a consequence of the higher ground temperatures. Horizontal wind
speed is found to be larger at Praia, also probably connected with the larger cloud-top heights. For
both Praia and Leipzig, a strong westerly-dominated wind direction is visible, while at Praia also an
easterly component is visible. It is interesting to note that at Praia mixed-phase clouds were mainly
observed in westerly airmasses.

The statistical parameters of the vertical-velocity distribution of each cloud case are shown in
Fig. 4 for both measurement sites. MEAN, STD, SKEW and KURT are derived from the vertical-
velocity values at cloud base measured with WiLi. An overview about the properties presented in
Figs. 3 and 4 is given in Tables 1 and 2 for Leipzig and Praia, respectively.

MEAN is distributed around $0\,\mathrm{m\,s^{-1}}$ for both sites. For STD, the distributions at Leipzig (Fig. 4c)
and Praia (Fig. 4d) show slightly different characteristics. At Leipzig a clear maximum at $0.4\,\mathrm{m\,s^{-1}}$
is visible, while at Praia the distribution of values seems to decay from small to larger values with
a maximum at $0.2\,\mathrm{m\,s^{-1}}$. However, the average of all STD values is equally $0.4\,\mathrm{m\,s^{-1}}$ for both





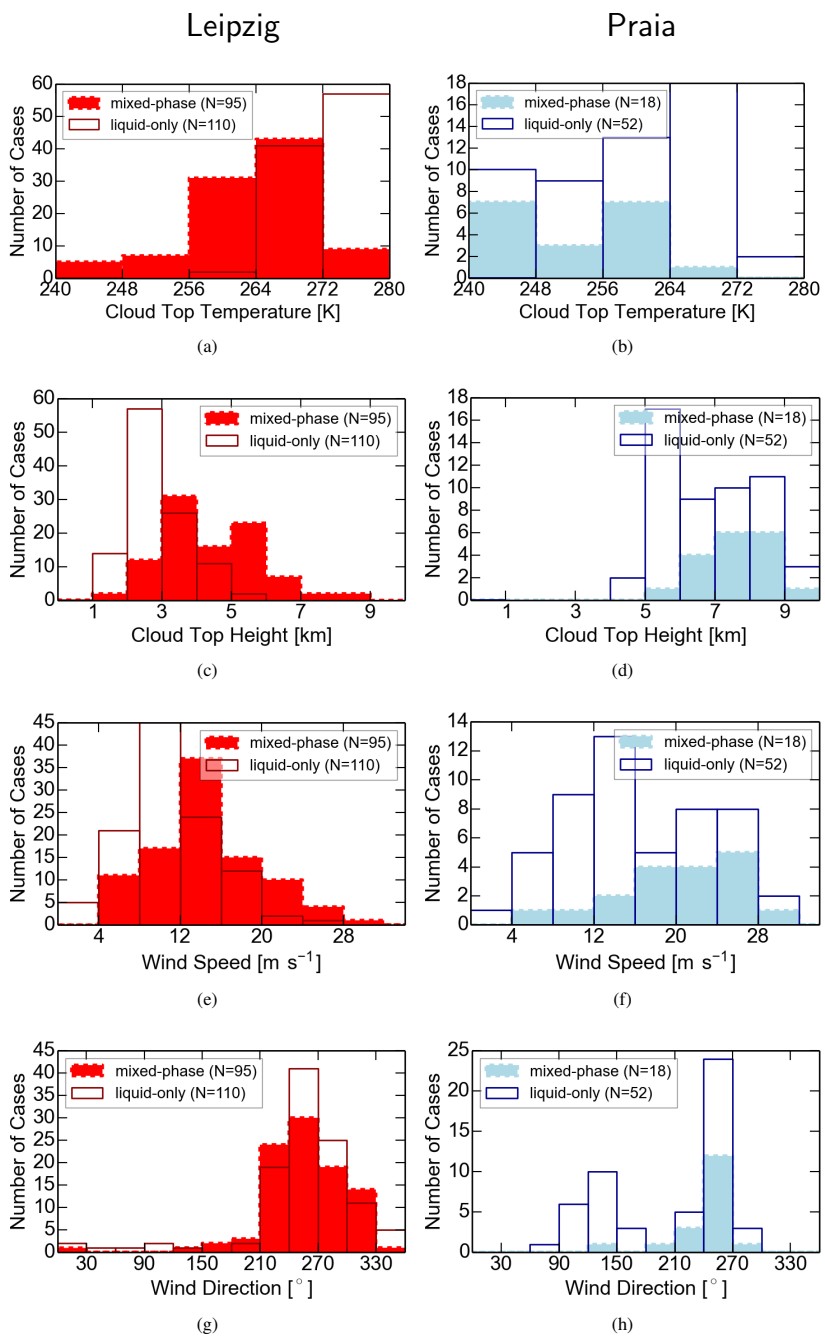

**Figure 3.** Histograms of temperature (a,b), height (c,d), horizontal wind speed (e,f) and meteorological wind direction (g,h) at cloud top of each cloud case measured over Leipzig (left column) and Praia (right column) separated into liquid-only (solid-line and white) and mixed-phase clouds (dashed and colored).



measurement sites. The larger maximum values of STD at Praia seem to compensate for the larger

amount of clouds with low STD. SKEW is mostly negative and on average $-0.3$ for Leipzig and

$-0.5$ for Praia. At Leipzig, mixed-phase clouds are mainly formed in layers where STD is between

$0.4$ and $0.5\,\mathrm{m\,s}^{-1}$. At Praia most mixed-phase clouds form under conditions with slightly lower

standard deviation of the vertical velocity. KURT is highly scattered for both sites, but only at Praia,

values $> 1.5$ seem to be possible. An effect that could be explained by a limited number of outliers

in the vertical velocity distribution at Praia.





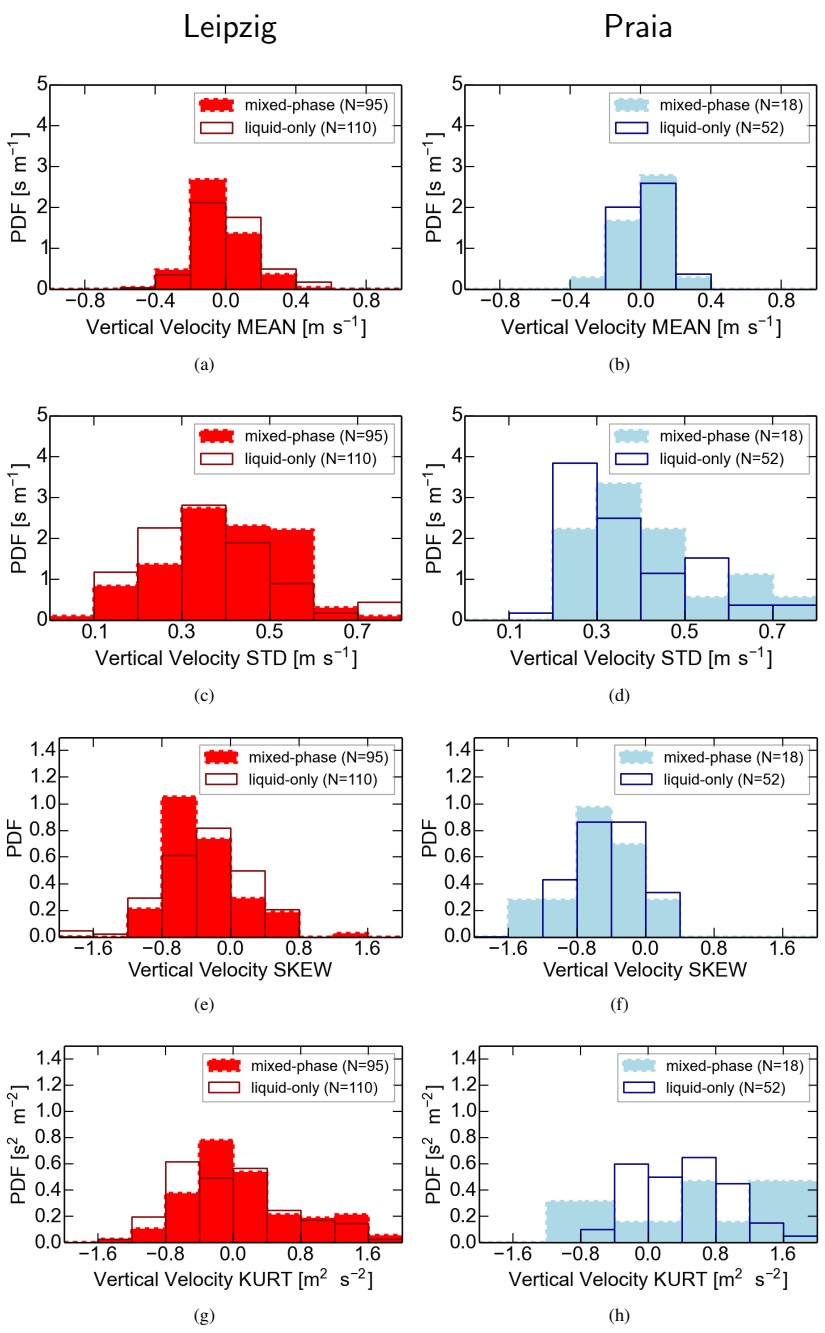

**Figure 4.** PDF of mean (a), standard-deviation (b), skewness (c) and kurtosis (d) of vertical velocity distribution for cloud cases at Leipzig (left column) and Praia (right column) separated into liquid-only (solid-line and white) and mixed-phase clouds (dashed line and colored).





**Table 1.** Overview about statistical parameters at Leipzig and their corresponding standard-deviation

| Parameter | Unit | Total | Liquid-only | Mixed-Phase |
|---|---|---|---|---|
| MEAN | $\mathrm{m\,s^{-1}}$ | $0.0 \pm 0.2$ | $0.0 \pm 0.2$ | $0.0 \pm 0.2$ |
| STD | $\mathrm{m\,s^{-1}}$ | $0.4 \pm 0.2$ | $0.4 \pm 0.1$ | $0.4 \pm 0.1$ |
| SKEW | None | $-0.3 \pm 0.5$ | $-0.3 \pm 0.4$ | $-0.3 \pm 0.4$ |
| KURT | $\mathrm{m^2\,s^{-2}}$ | $0.2 \pm 0.9$ | $0.2 \pm 0.7$ | $0.2 \pm 0.7$ |
| CTT | $^{\circ}\mathrm{C}$ | $269.0 \pm 7.7$ | $273.6 \pm 7.4$ | $263.7 \pm 7.4$ |
| CTH | km | $3.6 \pm 1.4$ | $2.9 \pm 1.4$ | $4.4 \pm 1.4$ |
| Wind dir. | $^{\circ}$ | $253.9 \pm 52.0$ | $253.1 \pm 44.2$ | $254.9 \pm 44.2$ |
| Wind vel. | $\mathrm{m\,s^{-1}}$ | $12.4 \pm 5.1$ | $10.7 \pm 5.1$ | $14.4 \pm 5.1$ |

**Table 2.** Overview about statistical parameters at Praia and their corresponding standard-deviation

| Parameter | Unit | Total | Liquid-only | Mixed-Phase |
|---|---|---|---|---|
| MEAN | $\mathrm{m\,s^{-1}}$ | $0.0 \pm 0.1$ | $0.0 \pm 0.1$ | $0.0 \pm 0.1$ |
| STD | $\mathrm{m\,s^{-1}}$ | $0.4 \pm 0.1$ | $0.4 \pm 0.1$ | $0.4 \pm 0.1$ |
| SKEW | None | $-0.5 \pm 0.4$ | $-0.4 \pm 0.5$ | $-0.5 \pm 0.5$ |
| KURT | $\mathrm{m^2\,s^{-2}}$ | $0.6 \pm 0.9$ | $0.5 \pm 1.2$ | $0.8 \pm 1.2$ |
| CTT | $^{\circ}\mathrm{C}$ | $257.4 \pm 9.2$ | $258.7 \pm 6.6$ | $253.3 \pm 6.6$ |
| CTH | km | $7.1 \pm 1.4$ | $6.9 \pm 1.0$ | $7.6 \pm 1.0$ |
| Wind dir. | $^{\circ}$ | $217.4 \pm 60.5$ | $207.9 \pm 33.4$ | $245.1 \pm 33.4$ |
| Wind vel. | $\mathrm{m\,s^{-1}}$ | $17.6 \pm 7.4$ | $16.8 \pm 5.9$ | $20.1 \pm 5.9$ |



### 4.2 Effect of time resolution on vertical-velocity statistics

In this section, the effect of different integration times on the Doppler-lidar derived vertical-velocity statistics is explored. The dataset at Leipzig was recorded on a grid of $2\,\mathrm{s}$ and is interpolated on 4, 6, 10 and $20\,\mathrm{s}$ in Fig. 5. STD and SKEW seem to be robust against an increasing interpolation time and change by less than $13\%$ when interpolation is increased from $2\,\mathrm{s}$ to $10\,\mathrm{s}$. For both values a change of less than $2\%$ is expected between the 4 and $5\,\mathrm{s}$ integration time. The values of KURT, however, seems to be more susceptible to longer integration times with an expected change of $25\%$ when changing from $4\,\mathrm{s}$ to $5\,\mathrm{s}$ integration time. The widths of the corresponding distributions of STD, SKEW and KURT do not change by more than $5\%$ and are, therefore, not shown here.

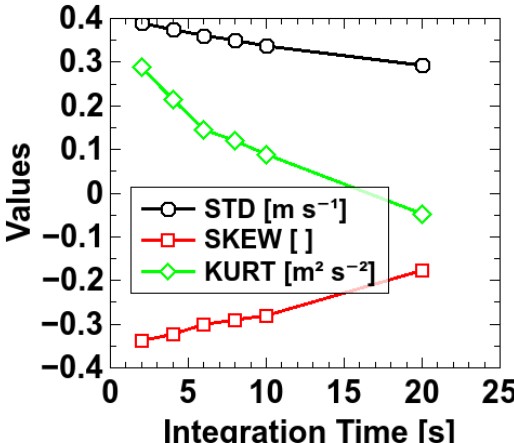

**Figure 5.** Mean values of STD, SKEW and KURT calculated for the Leipzig dataset for different interpolation times. The values at $4\,\mathrm{s}$ correspond to those shown in Tab. 1.

## 5 Discussion

The statistical properties of cloud-base vertical velocities shows a similar picture for both measurement sites. At Praia, the distribution of STD as presented in Fig. 4d is broader than that at Leipzig (Fig. 4c), but both distributions have the same mean value of $(0.4 \pm 0.1)\,\mathrm{m\,s^{-1}}$. The range of STD for cloud layers also agree to measurements of Fleishauer et al. (2002). For both sites, mixed-phase clouds show on average larger values of STD. There is an increase in probability for mixed-phase clouds at STD $> 0.5\,\mathrm{m\,s^{-1}}$. Taking into account all cases, the average increase of STD in mixed-phase clouds compared to liquid-only clouds is, however, below $0.1\,\mathrm{m\,s^{-1}}$.

Values of SKEW group around $-0.4$ for both sites, which indicates that in most of the cases radiative cooling at cloud top might be responsible for the turbulence within the cloud. However, it




must be kept in mind that values of SKEW $> 0$ are found at both sites. In these cases an external driving force like gravity waves or an influence of the radiative driven turbulent boundary layer is possible.

At Praia, vertical velocity KURT is on average larger by $0.5\,\mathrm{m^2\,s^{-2}}$. However, the large spread of the distribution of KURT of about $1\,\mathrm{m^2\,s^{-2}}$ at both measurement sites, makes this discrepancy less significant (see Fig. 4g and h). While MEAN, STD, and SKEW prove to be robust against changes in the measurement setup, KURT seems to be more strongly susceptible to changes in integration time. The way KURT is computed, it is very sensitive to outliers outside of the normal distribution. One explanation may be singular events that introduce strong vertical velocities into the clouds during a shorter time interval. Nevertheless, the on-average larger CTH at Praia could have also resulted in slightly different noise patterns in the observed vertical velocity fields, which could also explain the increased KURT at Praia.

All of the moment-based properties (STD, SKEW, KURT) behave similar for mixed-phase and liquid-only clouds, when compared at one measurement site. KURT shows larger extreme values but the statistical significance of this effect is low due to the small number of cases. Significant deviations occur for wind direction and wind speed. At both locations, mixed-phase clouds seem to have a higher probability of occurrence at wind speeds $> 15\,\mathrm{m\,s}$. At Praia, ice clouds seem to be strongly favored under westerly winds.

The original intention of the project was to characterize the behavior of vertical motions in cloud layers, admittedly expecting strong differences in the vertical motions between the tropics and the mid-latitudes. It is now very interesting to see, how well both statistics agree, given the fact that the two datasets were recorded in different climatic zones and some years apart. The duration of measurements at Praia was only eight weeks, while the dataset at Leipzig comprises nearly one year. At Praia, clouds appeared in groups with similar properties leading to the fact that statistical properties like STD also group for certain scenarios. In this way, peaks in the histograms at Praia can be explained. In spite of this, the overall similarity between both datasets is striking. This similarity hints that the physical process driving cloud layers may be the same in the mid-latitudes and the tropics. Once formed, cloud layers – mixed-phase or liquid-only – seem to be independent structures, kept alive by their internal vertical motions . The measurements presented in this work indicate that vertical velocities can not be responsible for the strong differences in the freezing-behavior of cloud layers between mid-latitudes and tropics found by Kanitz et al. (2011).

## 6 Summary and Conclusions

The statistics of vertical velocity at the bases of supercooled cloud layers has been analyzed on the basis of Doppler lidar measurements. The distribution of STD was found to be different at Praia with a surplus both at large and small values. Mean values of MEAN, STD, and SKEW were found to





agree within the statistical uncertainty. KURT was found to be larger at Praia. Significant differences were found in the advection speed of the airmasses, which is at Praia on average $4\,\mathrm{m\,s}^{-1}$ larger than at Leipzig. Also an absence of ice formation was found in clouds arriving at Praia from easterly direction.

The study was motivated by the question if regional differences in the vertical-velocity distribu-
240 tion in stratiform clouds can potentially contribute to the observed differences in the ice formation efficiency. From the observed similarity in the vertical-velocity statistics derived at the base of supercooled liquid cloud layers at the sub-tropical site of Praia and the mid-latitudinal site of Leipzig it can be concluded that other factors determine the efficiency of ice-formation processes. To date, it remains unclear which processes are responsible, but the amount of degrees of freedom reduces,
considering that humidity and temperature are constrained by the presence of (supercooled) liquid water and the limited depth of the analyzed shallow cloud layers.

*Acknowledgements.* The research leading to these results has received funding from the European Union Seventh Framework Programme (FP7/2007-2013) under grant agreement numbers 262254 (ACTRIS) and 603445 (BACCHUS) and from the HD(CP)$^2$ project (FKZ 01LK1209C and 01LK1212C) of the German Ministry for
Education and Research.



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
