# Peer review of "Statistics of vertical velocities in supercooled cloud layers over Leipzig and Praia measured with Doppler lidar"

_Atmospheric Chemistry and Physics, 2017_

## Referee Comment (RC1) · Anonymous Referee #3 · 19 Jun 2017

This paper described the vertical velocity statistics in supercooled shallow cloud layers for mixed-phase and liquid-only clouds over a sub-tropical site and a mid-latitudinal site based on ground base remote sensing. I appreciate the author analyzed statistics of vertical velocities over both sites, which is interesting for me, however this paper is not suitable for publishing in ACP for two reasons:

1. The main conclusion of this paper is "other factors besides cloud dynamics are responsible for the differences in ice formation efficiency reported previously for both sites." Also in the introduction the author posted the question "if this apparent effect of aerosols on ice formation could also be attributed to differences in vertical wind statis-

tics within mid-latitudinal and tropical cloud layers." I don't know why the author want to study the relationship between vertical wind statistics and ice formation. Based on classical nucleation theory, ice nucleation mainly depends on two variables temperature and ice nuclei efficiency. I don't know why ice nucleation should be related to vertical velocity statistics, and I didn't see any literature about this topic before. If this is "new physics", the author should address why they want to study this physically, not only just post a question without any base.

A more physically-based research topic for me is to study the source of ice nucleus. Figure 1 (adapted from Kanitz 2011) is very interesting for me, where at the same temperature, for example at -30, clouds at Praia and Punta show lower fraction of ice containing clouds than Leipzig. This study shows similar results as shown in Figure 3 a and b. Figure 3b shows that even at -33C (240 K), the fraction of ice containing clouds is still less than 50%. The only physically-based reason I can think about is that those liquid-only cloud don't have efficient ice nucleus. Because -33C is already a very low temperature for mixed phase cloud, as long as moderate efficient ice nucleus exists in those cloud (e.g., dust with freezing temperature around -25C) , ice nucleation would occurs. As far as I know, ice nucleation doesn't depend on vertical velocity statistics. It means that ice nucleation always occurs as long as supercooled water with dust inside stays at -33C, no matter whether it is in still environment, or in strong turbulence.

2. This paper compared several variables between two sites, including cloud top temperature, cloud top height, wind speed, wind direction, mean vertical velocity, STD, SKEW, KURT, however I didn't see enough discussions about the physics. For example, Figure 3h shows that more fraction of mixed-phase clouds over Praia when blowing westerly wind. So is it because the temperature is colder when blowing westerly, or is it lack of ice nucleus when blowing easterly?

---

## Referee Comment (RC2) · Anonymous Referee #1 · 20 Jun 2017

In the manuscript by Bühl et al, two remote sensing datasets are used to derive and compare the statistics of cloud base vertical velocities in supercooled and mixed-phase stratiform clouds at two different locations: In Leipzig and Cape Verde. Their results show that, for the observed cases, the vertical velocity statistics associated with the clouds at the two sites are very similar and attempt to couple this with the subject of ice formation efficiencies at the two sites.

Overall, datasets comprising detailed vertical velocity measurements from different regimes and especially in mixed-phase clouds, as shown in this manuscript, are interesting for the aerosol-cloud community. However, I am quite concerned about the

presentation of the analysis as well as the results and the conclusions, which I feel should be more substantial to be published in ACP and which would probably require changes beyond a major revision.

General comments

1. The Authors conclude that something else than cloud dynamics must affect the differences in ice formation, which clearly points towards the role of aerosols. In fact, the role of aerosols is acknowledged already in the Introduction, so I find it very limiting that this is not commented in the rest of the paper. I understand that the Authors would like to have this manuscript mainly focused on the vertical velocity statistics. However, given that the main conclusion is related to the cloud freezing efficiency and that the Authors limit their analysis to clouds where immersion freezing presumably dominates, I find this approach quite counter-intuitive and a serious concern for the scientific significance of the paper.

2. The Authors provide very little or no information about the differences in large-scale meteorological and environmental conditions, that can contribute to the cloud properties as well as the boundary layer mixing and thus vertical velocities. The analysis must go deeper and will need to include more details to provide the reader with more substantial information. It would be nice to also have explicit consideration of the boundary layer structure – whether the mixing is surface or cloud driven, or if the boundary-layer exhibits decoupled structure for example.

3. Since the analyzed data comprises relatively long periods of time especially for the Leipzig data, the previous point would probably call for analyzing also the seasonal differences – another interesting point which is not considered in the current manuscript. Do the same conclusions about the vertical velocity statistics hold throughout the year? Now the analysis comprises three kind of "random" months (Jan, Feb, June) at Praia vs. 3 full years from Leipzig lumped together.

Specific comments

1. Since the Authors focus on certain types of clouds and limit the temperature ranges etc in the sampling, should we actually expect to see large differences in the dynamics, i.e. vertical velocities? While the Authors state (Section 5, line 219) that they expected strong differences, it is not at all clear what sort of differences they are going after specifically. Please elaborate.

2. As stated in the manuscript, the ice formation can be expected to take place predominantly in the cloud top region, most likely through immersion freezing in this case, due to the screening of the cloud types. How would you rationalize the role of cloud base vertical velocities here, especially since immersion freezing essentially depends on the properties of the aerosol?

3. Section 5, line 207-209: Is this the case even when you limit the vertical velocity retrievals to -2...2 m/s?

---

## Author Comment (AC1) · 29 Jun 2017

Dear Referees and Editors,

We appreciate the unconcealed review provided for our manuscript. The main point of the critics points to the fact that we emphasize a potential relationship between vertical motion and ice formation. In fact, the manuscript is supposed to 'just' provide a study that compares regional differences in vertical air motions in mid-level stratiform clouds (which is barely available to date). We did not aim on providing full conclusions on the effect of the air motion on the microphysical properties of the cloud layer which might, however, be part of future studies. Similar studies have been presented, e.g. by Hill et

al. (2014) but they could only rely on very limited information about the actual vertical velocity statistics. This why we think that it is important to show the basic statistics of vertical motions in mixed-phase cloud layers.

We also want to make clear that we do not suspect a potential (and theoretically unjustified) relationship between ice nucleation and vertical air motions. We used the term "ice formation" to identify the complete process from ice nucleation via ice particle growth to sedimentation of the ice particles which has been shown to be strongly dependent on vertical motions, e.g., by Korolev and Field (2008). We can only suspect that the title of the UDINE study (Up- and Downdrafts in drop and Ice Nucleation Experiment) might have been misleading in this context.

We thus suggest that we will modify the manuscript by means of a major revision: We will better define with the term "ice formation" and straighten our motivation to the question about general comparability of vertical air motion in tropical and midlatitudinal mid-level clouds. The updated introduction will contain a condensed review of the potential effects of vertical air motion on cloud microphysical properties. The conclusions will concentrate on the found similarity in the vertical velocity distributions that is actually shown for the first time.

Yours sincerely,

Johannes Bühl and coauthors.

References used in this reply:

Hill, A. A., Field, P. R., Furtado, K., Korolev, A. and Shipway, B. J. (2014), Mixed-phase clouds in a turbulent environment. Part 1: Large-eddy simulation experiments. Q.J.R. Meteorol. Soc., 140: 855–869. https://doi.org/10.1002/qj.2177

Korolev, A. and Field, P.R. (2008), The Effect of Dynamics on Mixed-Phase Clouds: Theoretical Considerations. J. Atmos. Sci., 65, 66–86, https://doi.org/10.1175/2007JAS2355.1

---

## Referee Comment (RC3) · Anonymous Referee #3 · 30 Jun 2017

Dear Johannes Bühl and coauthors,

I agree that the vertical velocity statistics is worth to investigate. Effect of vertical air motion on cloud microphysical properties is a hot and challenging research topic nowadays, much better than the effect of vertical velocity on "ice formation". You had valuable data and also did hard work on data analysis. Please be careful what scientific question you want to address in your revised version.

Some comments on the two papers you mentioned about:

Korolev and Field: If I understand correctly, they are interested in the activation of liquid cloud droplets in an ice cloud parcel. Sedimentation which is sensitive to vertical

velocity is ignored in their paper. Their research interest is quite different than what you want to investigate in Fig.1 (your paper). Their conclusion is that strong vertical velocity or fluctuation can active cloud droplets in ice phase cloud, thus lead to the formation of mixed phase cloud. However Fig. 1 shows the fraction of ice containing clouds is quite different at two sites, and especially low at Praia and Punta Arenas. The main difference is that in their case, the cloud always contains ice, while in your case fraction of ice containing cloud is very low. If their hypothesis is true, the mixed phase cloud is all formed through the activation of liquid cloud droplets in ice cloud, fraction of ice containing clouds should be 1.

Hill et al.: This paper is actually using LES to test Korolev's hypothesis. Because it is designed to test it, the basic setup is suitable for the theory. For example, they "prescribed the ice number concentration throughout the domain and an ice mass mixing ratio of 0.1 g kg-1. In the base simulation, ice is not permitted to sediment or growth in size, but ice mass and number are advected by the wind." Later, they allow ice to sediment. However, "an ice source is required so that sedimentation does not deplete all ice from the domain. . . at the top of domain ice mass and number are fixed so they are equal to the initial values. This provides a constant source of falling ice into the domain, which maintains the background ice number concentration and mass mixing ratio at approximately the same values as those in the base simulations." Therefore, in this paper, the liquid water cloud fraction is less than 1, but the ice containing cloud fraction is always 1, which is quite different than Fig.1 in your paper.

Best,
* * *

---

## Referee Comment (RC4) · Anonymous Referee #1 · 2 Jul 2017

Dear Johannes Bühl et al.,

The suggestion of a major review to clarify the objectives and scope of the paper does sound reasonable to me, though I would think it will be quite a lot of work. Again, I would also still hope that the analysis and the interpretation of the physics behind the observed vertical velocity statistics would go deeper. A little more work there would provide the readers much better insight in this interesting topic and in general result in a much stronger manuscript.

Regards,

---

## Editor Comment (EC1) · M. Tesche (Editor) · 3 Jul 2017

Dear Johannes,

both Referees support your proposal of sharpening the aim of your work. They also agree to change their initial recommendation to major revisions based on your suggested changes to the manuscript. Please account for the Referees' comments and suggestions when revising your work.

With best regards,

Matthias